# A Deep Learning-Based Multi-Signal Radio Spectrum Monitoring Method for UAV Communication

**Changbo Hou** [1,2] , **Dingyi Fu** [2,*] , **Zhichao Zhou** [2] **and Xiangyu Wu** [2]

1   Key Laboratory of Advanced Marine Communication and Information Technology,
    Ministry of Industry and Information Technology, Harbin Engineering University, Harbin 150001, China;
    houchangbo@hrbeu.edu.cn
2   College of Information and Communication Engineering, Harbin Engineering University,
    Harbin 150001, China; zhouzhichao@hrbeu.edu.cn (Z.Z.); wuxiangyu@hrbeu.edu.cn (X.W.)
*   Correspondence: fdy_0429@hrbeu.edu.cn

**Abstract:** Unmanned aerial vehicles (UAVs), relying on wireless communication, are inevitably influenced by the complex electromagnetic environment, attributed to the development of wireless communication technology. The modulation information of signals can assist in identifying device information and interference in the environment, which is significant for UAV communication environment monitoring. Therefore, in scenarios involving the communication of UAVs, it is necessary to find out how to perform the spectrum monitoring method to obtain the modulation information. Most existing methods are unsuitable for scenarios where multiple signals appear in the same spectrum sequence or do not use an end-to-end structure. Firstly, we established a spectrum dataset to simulate the UAV communication environment and developed a label method. Then, detection networks were employed to extract the presence and location information of signals in the spectrum. Finally, decision-level fusion was used to combine the output results of multiple nodes. Five modulation types, including ASK, FSK, 16QAM, DSB-SC, and SSB, were used to simulate different signal sources in the communication environment. Accuracy, recall, and F1 score were used as evaluation metrics. The networks were tested at different signal-to-noise ratios (SNRs). Among the different modulation types, FSK exhibits the most stable recognition performance across different models. The proposed method is of great significance for wireless radio spectrum monitoring in complex electromagnetic environments and is adaptable to scenarios where multiple receivers are used in vast terrains, providing a deep learning-based approach to radio monitoring solutions for UAV communication.

**Keywords:** radio monitoring; UAV communications; deep learning; automatic modulation recognition; multi-signal recognition; decision-level fusion



## 1. Introduction

With the rapid development of unmanned aerial vehicles (UAVs), applications in various fields have experienced explosive growth. As high-mobility platforms based on wireless communication, UAVs are closely associated with wireless communications and communication environments. However, the current communication environment has become increasingly complex, posing numerous challenges to the communication effectiveness, security, and robustness of UAVs [1–4].

Driven by the development in the fields of communication, electronics, and computers, coupled with the proliferation of wireless devices, the wireless channel environment has become highly complex [5]. The spectrum has become congested and fraught with threats. Consequently, the communication environment for UAVs is significantly affected. The current wireless communication spectrum is extensively utilized by various wireless devices, including mobile communications, wireless local area networks, and Bluetooth devices,

resulting in limited and fiercely contested spectrum resources. This spectrum congestion and interference intensify the difficulty of UAV communication, potentially leading to degraded communication quality or even interruptions. Moreover, UAVs face challenges in wireless communication security. With the widespread use of UAVs, security risks such as unauthorized UAVs entering restricted areas or UAV hijacking have emerged [6–8]. These threats impose new requirements on the confidentiality and reliability of UAV communication, necessitating effective security measures to protect UAV communication data and systems.

To overcome the impact of the current complex communication environment on UAV communication, spectrum analysis, monitoring, and management are essential. These measures will facilitate efficient spectrum planning and allocation, enhancing the reliability and performance of UAV communication. Additionally, spectrum monitoring helps to strengthen the security of UAV communication, including the monitoring of unknown communication devices and potential communication interference signals, which ensures secure UAV operations and data transmission.

For radio monitoring, modulation recognition is the initial and crucial step. Obtaining the modulation information of signals allows for demodulation and further analysis. The modulation information itself plays a significant role in identifying the identities of communication devices and assessing communication interference in the current environment. Therefore, modulation recognition is necessary for radio monitoring.

### 1.1. Related Work

In the analysis and monitoring of non-cooperative wireless communication environments lacking prior knowledge, automatic modulation recognition (AMR) technology is indispensable [9–11]. Modulation identification information is essential to demodulate signals and allow further analysis. Without the application of AMR, modulation recognition is heavily based on manual judgment or more complex demodulators. These methods are difficult to guarantee accuracy, and the implementation of the system becomes very difficult in a straightforward manner.

Currently, there have been numerous studies on AMR technology for the recognition of individual signals. In classical approaches, there are mainly two methods: likelihood-based methods and feature-based methods [12]. In [13], Zhang et al. proposed a likelihood ratio-based AMR method. This method can identify orthogonal frequency division multiplexing (OFDM) with index modulation signals and discusses both known channel state information and blind recognition scenarios. Ghauri et al. [14] introduced a classifier for pulse amplified modulation (PAM) and quadrature amplified modulation (QAM) signals. The classifier is based on hidden Markov models (HMM) and genetic algorithms, achieving an accuracy of 88% under SNR is 0dB and a signal sampling point count of 1024. Punith Kumar H. L. et al. [15] utilized a feature-based pattern recognition method, employing only three crucial features to achieve a good recognition rate, even at low SNR. At SNR = 4 dB, it achieved an overall recognition rate of 98.8% for six digital modulation signals. M. Abu-Romoh et al. [16] proposed a hybrid automatic modulation classification method, which lies between likelihood-based and feature-based classifiers, relying on statistical moments along with a maximum likelihood engine. This classifier achieved 100% accuracy in classifying QAM and PSK at 18 dB but exhibited a significant decrease in accuracy at low SNR. In recent years, deep learning (DL) methods have garnered attention from researchers due to their excellent classification performance, and there has been a lot of research work based on DL [17–20]. Y. Wang et al. [21] presented a constellation-based convolutional neural network (CNN) ensemble consisting of two CNNs. The first CNN layer, named DrCNN, is responsible for recognizing BPSK, QPSK, QAMs, GFSK, CPFSK, and PAM4. The second CNN layer, named MaxCNN, focuses on recognizing QAMs within the range of −8 dB to 18 dB, with 2 dB intervals. The recognized signal types include BPSK, QPSK, 16QAM, 64QAM, GFSK, CPFSK, and PAM4. When the SNR exceeds 4 dB, DrCNN achieves an accuracy of over 95%, outperforming RNN, DNN, and Inception models for

such tasks. MaxCNN achieves an accuracy close to 100% in distinguishing between 16QAM and 64QAM. It can be observed that extensive research has been conducted on AMR of individual communication signals in the past.

And many related research works also exist on using AMR for radio monitoring in the UAV communication environment [22]. Emad et al. [23] proposed a method based on deep learning for radio frequency (RF) signal spectrum monitoring and deployed the neural network on FPGA. For UAV radar signal modulation recognition, Liu et al. [24] proposed a method based on bispectral slice, which performs excellently under a low SNR. Compared with most modulation identification methods, these methods use more digital and radar signals and are suitable for radio monitoring in the UAV communication environment. However, most existing works also do not consider the presence of multiple signals on the spectrum to be identified.

In a diverse communication environment, the probability of multiple non-cooperative wireless devices appearing simultaneously increases. In order to effectively monitor the spectrum within a certain frequency range, it is necessary to simultaneously identify and analyze multiple wireless signals captured. However, the majority of AMR methods typically only work for a single signal. Therefore, improvements to AMR methods for multiple signals are necessary. Hou et al. [25] proposed a sliding window-based spectrum segmentation method that divides the spectrum into smaller regions containing the signals of interest, and then applies AMR method for the individual signal to each of them. This method achieves high detection accuracy, but the entire detection process is not end to end, leading to a reduction in detection speed. Due to the rapid changes in the wireless communication environment, real-time capability is crucial for wireless monitoring. Table 1 evaluates the related work.

**Table 1.** Related works.

| Related Works | Method | Strength | Weakness |
|---|---|---|---|
| Zheng et al. [13] | Likelihood-Based | Both known channel state information and blind recognition scenarios are discussed. | Restricted to OFDM. |
| Ghauri et al. [14] | HMM and genetic algorithms | Higher recognition accuracy than other traditional methods. | Few types of recognition, and only one can exist at the same time. |
| Punith Kumar H.L et al. [15] | Decision theoretic approach | Based on the minimum feature extraction, quickly performed through the decision tree. | Compared with the accuracy of using a more complex deep learning model, the recognition accuracy is still not high; still limited to only one signal at one time. |
| M. Abu-Romoh et al. [16] | likelihood-based and feature-based | Achieved 100% accuracy in classifying QAM and PSK at 18dB. | Exhibited a significant decrease in accuracy at low SNR; still limited to only one signal at one time. |
| Y. Wang et al. [21] | CNN (DrCNN and MaxCNN) | Various digital communication signals including 16QAM and 64QAM can be distinguished. | Requires multiple data set inputs; only one signal can be resolved in data at a time. |
| Emad et al. [23] | CNN | Have been implemented on a GPU and an FPGA; modulation types up to 11 kinds. | Recognition accuracy is slightly lower; still only single signal modulation recognition. |
| Liu et al. [24] | GA-BP neural network | Good performance on radar signals recognition under the low SNR conditions. | Common communication modulations are not covered and still only for single signal modulation recognition. |
| Hou et al. [25] | Complex-ResNet and sliding window | High detection accuracy, multi-signal covered. | Not end-to-end, leading to a reduction in detection speed. |

Therefore, it is important to explore a multi-signal AMR method that offers fast detection speed and a shorter detection process while ensuring minimal fluctuations in accuracy.

Additionally, single-node spectrum monitoring is significantly constrained in environments with specific monitoring range requirements or in the presence of severe shadowing or fading effects [26,27]. The introduction of multiple receiver nodes working collaboratively can effectively enhance the radio monitoring capability in demanding environments.

*1.2. Contributions*

Currently, individual signal AMR is insufficient to support the monitoring of complex communication environments and lacks a multi-signal AMR method with a shorter processing time. This paper proposes a deep learning-based multi-signal recognition method that achieves the recognition of coexisting signals in the frequency domain that are separable and temporally overlapped. Firstly, a multi-signal spectrum dataset is constructed, including signal categories such as M-ASK, FSK, 16QAM, AM, and AM-SSB. Each spectrum file contains 1 to 4 signals with SNR ranging from −8 dB to 6 dB with 2 dB intervals. Subsequently, three different architectures of multi-signal recognition networks are built, and the advantages and disadvantages of these three networks are analyzed. Additionally, a decision fusion method is introduced in the experiments to improve the accuracy of the proposed method. Finally, the performance of the method is tested, including the recall, precision, and F1-score. The innovations of this work are detailed as follows:

- A method of multi-signal modulation recognition based on a one-dimensional neural network is proposed. The network structure is relatively simple, and multiple communication signals can be considered.
- A multi-node joint decision-making model is considered under a distributed architecture. Only the decision results of each node need to be transmitted for fusion, which effectively reduces the data transmission cost. The method requires little calculation and can quickly detect whether a signal is in the target frequency band. Thus, applying this method in nodes will not incur excessive computational pressure and delays.

The remainder of this paper is organized as follows. Section 2 introduces the system architecture. Section 3 presents the implementation method of multi-signal recognition based on the one-dimensional neural network, including the spectrum dataset, neural network structure, training and testing methods, and the decision fusion method. Section 4 analyzes the performance and Section 5 concludes the paper.

## 2. System Architecture

*2.1. Multi-Signal Spectrum Dataset*

Assuming that there are $M$ receiver nodes in total and $N$ unknown signals to be recognized in the UAV communication spectrum in the same period, after collecting information via hardware, the time domain information is determined as follows:

$$x_j(t) = \sum_{i=0}^{N-1} u_{ji}(t) + n_j(t), \tag{1}$$

where $u_{ji}(t)$ is the $i$-th unknown signal to be detected in the spectrum sensed by the $j$-th receiver node, and $n_j(t)$ is the additive white Gaussian noise (AWGN) in the channel where the $j$-th receiver node is located. It is obviously difficult to extract effective information in the time domain. After fast Fourier transform (FFT), the frequency domain is denoted as follows:

$$X_j(\omega) = \sum_{i=0}^{N-1} U_{ji}(\omega) + N_j(\omega), \tag{2}$$

where $U_{ji}(\omega)$ does not overlap in the same receiver node.

The position of the spectrum and the occupied bandwidth of $U_i(\omega)$ are limited, and SNR varies from $-8$ to 6 dB in 2 dB intervals. The selected signal types cover digital and analog modulation schemes, and the number of points in each segment of the wideband spectrum data is 1024. The specific parameters are shown in Table 2, where $f_s$ is the sampling frequency.

**Table 2.** Parameters for the multi-signal spectrum dataset.

| Type | Parameters | Range |
|---|---|---|
| M-ASK | Carrier Frequency $f_0$ | (0.07–0.43)*$f_s$ |
| | Symbol Width $T_b$ | (1/25–1/10)*N/$f_s$ |
| | $M$ | [10,15,20,25] |
| 2FSK | Carrier Frequency $f_1$, $f_2$ | (0.07–0.43)*$f_s$, |
| | | $0.04 \leq \dfrac{|f_1 - f_2|}{f_s} \leq 0.08$ |
| | $T_b$ | (1/25–1/10)*N/$f_s$ |
| 16QAM | $T_b$ | (1/25–1/10)*N/$f_s$ |
| DSB-SC | $f_0$ | (0.07–0.43)*$f_s$ |
| | Baseband $f_h$ | (0.005–0.007)*$f_s$ |
| SSB | $f_0$ | (0.07–0.43)*$f_s$ |
| | $f_h$ | (0.005–0.007)*$f_s$ |

We employ text format .txt files to label and store the dataset. The tags contain the types, occupied bandwidth, and positions of the different communication signals.

The original training set has a total of 32,000 data. These data are organized into the structure that is shown in Figure 1. The training set can also be split into multiple subsets for specific applications.

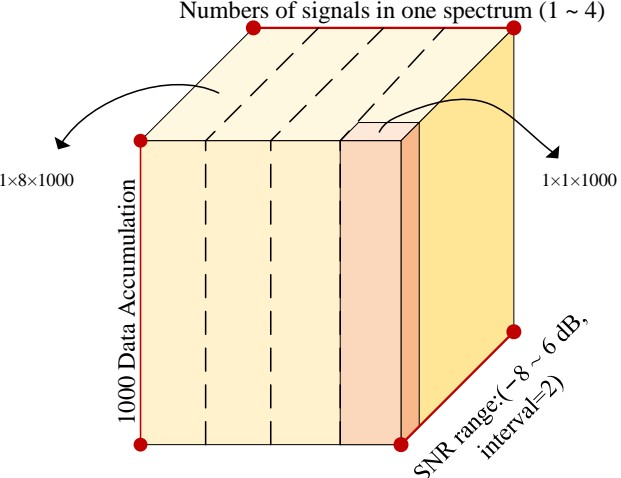

**Figure 1.** Dataset structure.

Different datasets are generated according to the needs of different test methods to create the test sets. The dataset used for each test will be described before the corresponding experiment result.

### 2.2. Processing in the Single Node

Before the fusion, the corresponding single node must perform spectrum detection for the UAV communication environment. There may be multiple communication devices in the environment, and we may also have multiple detection nodes. The specific flow chart is shown in Figure 2.

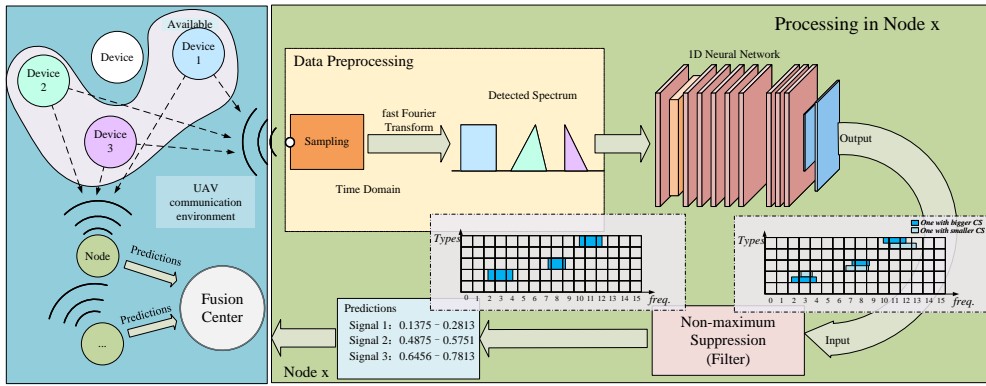

**Figure 2.** Single-node spectrum sensing process.

The concept of our proposed method originates from object detection in image processing. Firstly, data preprocessing is carried out. The data are converted from the time domain to the frequency domain by fast Fourier transform (FFT). After that, the spectrum data are input into a one-dimensional neural network to sense the frequency domain information, including detecting the existence of signals and the occupied frequency bandwidth and positions. The frequency band information must be normalized when inputting. After processing, some predicted positions and modulation labels are output. The redundant predictions are then cleaned up by the non-maximum suppression algorithm. Finally, the label of each signal is obtained, which realizes the frequency bandwidth occupation and the frequency band position detection of the signals. The one-dimensional neural network is an essential component in multi-signal modulation recognition. The specific structure is explained in Section 3.

### 2.3. Multi-Node Fusion Process Structure

The method in this paper adopts a centralized structure, which is shown in Figure 3. Multiple nodes aggregate the data to the fusion center, which then makes the final decision. Different nodes occupy different frequency bands and positions, which means the interferences are not the same, and there are subtle differences between models in nodes. Each node then gives its own prediction under a different environment, and these are returned to the fusion center. Finally, the fusion center employs decision fusion to vote on the different results reported by multiple nodes.

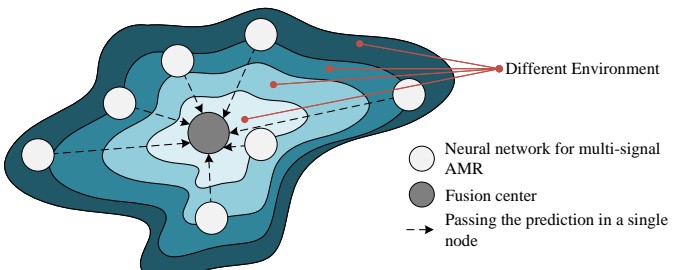

**Figure 3.** Multiple nodes environment.

## 3. Proposed Approach

### 3.1. Using Neural Networks in Multi-Signal AMR

Machine learning technology has applications in many fields, especially deep learning. In the field of wireless communication, deep learning technology can respond quickly, without prior knowledge, adapt to rapidly changing massive data, and has many applications in signal recognition and processing [28–30]. It is already being employed in wireless device identification [31]. Deep learning models can easily complete classification or detection tasks, provided there is sufficient data support [32–34]. As the problem of multi-signal

AMR is transformed into the detection of the frequency bandwidth and location occupied by the multiple signals in the spectrum, the concepts and ideas of deep learning in image processing and object detection can also be borrowed.

Analogous to object detection, either a one-stage or two-stage scheme can be used for multi-signal AMR. In a one-stage scheme, the input data are divided into a limited number of cells; if the center position occupied by the signal is in a certain cell, then this cell is responsible for detecting it. A two-stage scheme employs a sliding window like [25] or a region proposal method without inputting the entire spectrum into the network. Although the two-stage scheme has higher accuracy, the detection speed is relatively low. As the spectrum monitoring method requires good real-time performance, we employ the one-stage scheme in this work, sacrificing a small amount of accuracy in exchange for a significantly improved detection speed. Figure 4 shows the structure of the one-dimensional neural network for multi-signal AMR. The first area in Figure 4 is the feature extraction part, which can be arbitrary as long as the network structure can realize the feature extraction. In Figure 4, the convolutional neural network (CNN) is employed as an example.

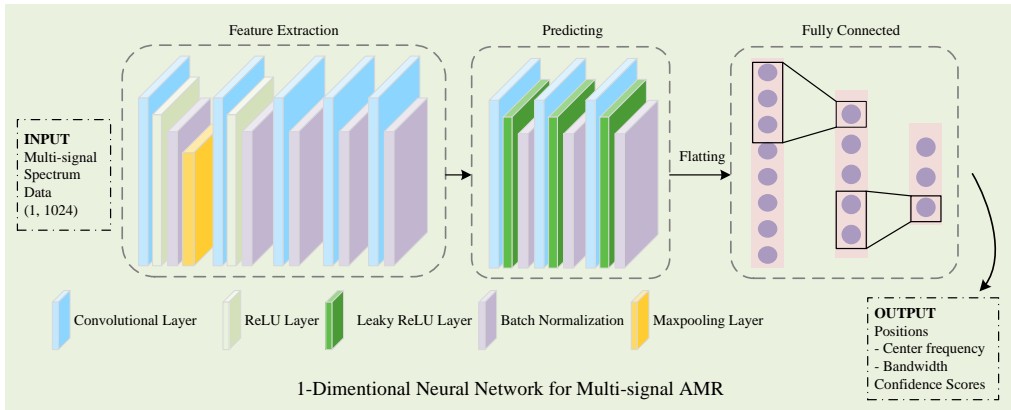

**Figure 4.** One-dimensional neural network for multi-signal AMR.

This paper uses three feature extraction networks: CNN, convolutional long-short-term memory fully connected deep neural network (CLDNN), and deep complex convolutional network (complex-conv). CNN is one of the most commonly used networks for image processing and classification, which employs a stack of convolutional and pooling layers to extract features. CNN models have a small size and short training time. CLDNN is a composite network that uses CNN, long short-term memory (LSTM), and deep neural network (DNN) structures simultaneously. Because of the LSTM cell, CLDNN is reasonably effective at dealing with sequential data. Adding the CNN before LSTM reduces the variance and dimension of the sequence data first, which is followed by the DNN structure for nonlinear mapping. The three models have complementary advantages and significantly improve the performance of sequence data processing. Trabelsi et al. [35] proposed complex-conv with richer expressiveness and data processing capabilities in 2017. Experiments were carried out to test the complex-conv models on several computer vision and music transcription tasks using the MusicNet dataset, achieving state-of-the-art performance. In this paper, complex-conv is mainly implemented using complex convolutional layers and complex pooling layers. The three extraction networks have their unique features.

The network will require more parameters and a longer training time as the expressive ability improves. Table 3 shows the parameters of the one-dimensional neural network for recognition built with different feature extraction networks and illustrates that complex-conv has more parameters than the other two networks.

**Table 3.** Total parameters in three networks for recognition.

| Feature Extraction Parts | CNN | CLDNN | Complex-Conv |
|---|---|---|---|
| Total Parameters | 39.0 M | 41.1 M | 77.3 M |

### 3.2. Data Preprocessing for the Dataset

The initial time-domain data is first generated according to the dataset parameters shown in Table 2. The FFT is then employed to transfer the information to the frequency domain, and the training data are labeled by category, center frequency, and occupied bandwidth. Each piece of spectrum data is divided into 16 subsections, with 64 points per section. Each subsection contains two prediction data for the signal whose center frequency is in that subsection, including center frequencies, occupied bandwidth, confidence scores (CSs), and modulation classifications. The specific composition is shown in Figure 5. The length of the modulation classification possibilities is five because the current dataset is divided into five categories according to the preset modulation method used in the communication, in the order of ASK, FSK, QAM, DSB, and SSB. If the ground truth value is used during training, the possibility for the actual category is marked as 1, and the others are 0. For example, if the communication signal is FSK, then the tensor will be 01000. The network's autonomous judgment of the category provides the predictions. If there is no signal in this section, the data in this tensor is all 0. In this way, a wideband spectrum data label can be converted into a $16 \times 11$-dimensional tensor.

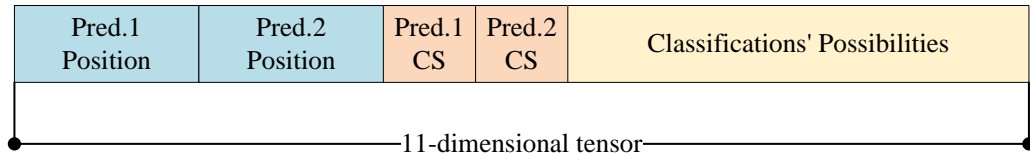

**Figure 5.** The 11-dimensional tensor for each section.

### 3.3. Training and Testing Method

We perform two stages of training in our network. Because the network structure is too deep, the integrated training of the network may cause the convergence rate to be too slow or in a state of nonconvergence. The first stage trains the feature extraction network. We observe that in many classical deep learning networks, the structure of the classification network can effectively extract features from data. Therefore, the framework of classification networks is employed to train a network for extracting features from spectrum data. The second stage is the overall training. The trained feature extraction network parameters are then employed as the initialization parameters of the recognition network, which significantly reduces the training time. Algorithm 1 is the overview of the training procedure.

#### 3.3.1. Pretraining of the Feature Extraction Network

After adding a flattening layer and a fully connected layer after the feature extraction network, the model is trained as a multi-class network. The parameters in Table 2 are employed to generate a new individual signal dataset for training. We generate 4000 spectrum data for each category and use 10% as the validation set. The model is saved after training the fixed epochs. Three multi-classification networks require training at this point: CNN, CLDNN, and complex-conv. When training, the cross-entropy function is used as the loss function.

#### 3.3.2. Overall Training

First, the network parameters saved in the previous stage are correspondingly attached to the recognition network. It should be noted that the layer names here must be consistent with the multi-class network layer names. The previously generated dataset is then employed for training, in which the output results are 176-dimensional tensors. The

model is saved after training for a fixed number of epochs. When training overall, the loss function is different from the first stage. The following section describes the loss function used in the overall training.

---

**Algorithm 1** Training the recognition networks

---

**Input:** Feature Extraction Dataset (or Subset) $D_1$, Multi-signal Dataset (or Subset) $D_2$;
**Output:** Neural Network Parameters $\theta_s$
1:  TRAIN_FEN($D_1$) :
2:      $\theta_{\mathrm{FE}} \leftarrow$ **initialize randomly**
3:      **for** e $\in$ Maxepoch$_1$ **do**:
4:          $(x, y) \leftarrow$ random mini-batch from $D_1$
5:          $\hat{y} \leftarrow$ forward-propagation($x$, $\theta_{\mathrm{FE}}$)
6:          $e_1 \leftarrow$ cross-entropy($y$, $\hat{y}$)
7:          $\theta_{\mathrm{FE}} \leftarrow$ backpropagation($e_1$)
8:      **end**
9:      **return** $\theta_{\mathrm{FE}}$
10: TRAIN_SN($\theta_{\mathrm{FE}}$, $D_2$) :
11:     $\theta_{\mathrm{S}} \leftarrow$ **initialize randomly**
12:     $\theta_{\mathrm{S}} \leftarrow$ **corresponding layer parameters**($\theta_{\mathrm{FE}}$)
13:     **for** e $\in$ Maxepoch$_2$ **do**:
14:         $(x, y) \leftarrow$ random mini-batch from $D_2$
15:         $\hat{y} \leftarrow$ forward-propagation($x$, $\theta_{\mathrm{S}}$)
16:         $e_2 \leftarrow$ cross-entropy($y$, $\hat{y}$)
17:         $\theta_{\mathrm{S}} \leftarrow$ backpropagation($e_2$)
18:     **end**
19:     **return** $\theta_{\mathrm{S}}$

---

3.3.3. Loss Function for Recognition Networks

Because supervised learning is the mapping relationship between the raw data and the label in the fitting sample, we need to design a function to estimate the loss between the predictions and the ground truth. Observing the loss will indicate the quality of the network parameters fitting. Three quantities need to be considered in the recognition network: localization, confidence scores, and modulation categories. Therefore, the error used for the overall network is also divided into three parts, namely, the localization, confidence, and classification errors. The confidence error is divided into errors that are with or without a signal:

$$e_1 = \lambda_1 \sum_{i=0}^{a} \sum_{j=0}^{b} \mathbb{1}_{ij}^{sig} \left[ (x_i - \hat{x}_i)^2 + \left( \sqrt{w_i} - \sqrt{\hat{w}_i} \right)^2 \right],$$

$$e_2 = \sum_{i=0}^{a} \sum_{j=0}^{b} \left( \mathbb{1}_{ij}^{sig} + \lambda_2 \mathbb{1}_{ij}^{nosig} \right) (c_i - \hat{c}_i)^2, \tag{3}$$

$$e_3 = \sum_{i=0}^{a} \mathbb{1}_{i}^{sig} \sum_{c \in \text{classes}} (p_i(c) - \hat{p}_i(c))^2.$$

The sum of $e_1$, $e_2$, and $e_3$ is the entire loss function. In Equation (3), $e_1$ is the localization error, including the center frequency error and the occupied width error. We square the widths to balance the errors of different occupied widths, which improves the identification accuracy of signals with narrower occupied bandwidths. Additionally, $e_2$ is the confidence error, where $c_i$ is the prediction confidence score, which is calculated as $\mathrm{Pr}(s) \times \mathrm{IoU}_{pred}^{truth}$. If the signal exists in a section, $\mathrm{Pr}(s) = 1$, and the prediction confidence score will be the intersection over union (IoU) of the prediction and the ground truth. If a signal does not exist, it will be zero. Finally, $e_3$ is the classification error.

Of the three errors, $\lambda_1$ and $\lambda_2$ are weights of different parts. In this paper, $\lambda_1$ and $\lambda_2$ are taken as 5 and 0.5, respectively; *a* is the number of sections of spectrum data; *b* is the number of intra-section signal predictions for each section. We take 16 and 2, respectively. Here, $1_{ij}^{sig}$ indicates that a signal exists in the *j*-th prediction of section *i* and $1_{ij}^{nosig}$ indicates that no signal exists; $1_{ij}^{sig}$ and $1_{ij}^{nosig}$ are complementary; and $1_i^{sig}$ means a signal exists in the section *i*.

### 3.3.4. Testing Method

Before testing, the test set is generated according to the parameters in Table 3, and then the categories and positions of signals are predicted through one-dimensional neural networks. The category with the highest possibility is selected as the signal's category in each section, and then NMS is used to select the final output of the network.

First, the predictions in the same category are sorted in one section according to the CS, where the highest prediction is taken as the final output prediction. The IoU of the other prediction is then calculated, where the result with the highest CS is taken. All predictions above a certain threshold are deleted, and the process is repeated until all redundancies are eliminated.

### 3.3.5. Evaluation Method

Corresponding evaluation indicators are required to evaluate the quality of a network. In this paper, we employ the precision, recall, F1-score, and P-R curve as performance indicators. We cannot derive the true negative $TN$ values in a section of spectrum data, as there is no set minimum separation distance between signals, so we cannot use accuracy as an evaluation metric. In this paper, the true positive $TP$, false positive $FP$, and missing detection $FN$ values are well judged. When the IoU of the predicted value and the ground truth value is greater than 0.65, we regard it as a $TP$. When the IoU between the prediction and the ground truth value is less than or equal to 0.65, or when the signal is not reflected in the prediction, we regard it as an $FN$. When a signal detected in the prediction is not represented in the ground truth, we treat it as an $FP$. The specific expressions of precision and recall are shown in Equation (4):

$$
\begin{aligned}
\text{Precision} &= \frac{TP}{TP + FP}, \\
\text{Recall} &= \frac{TP}{TP + FN},
\end{aligned}
\tag{4}
$$

We then calculate the F1-score with the known precision and recall using Equation (5):

$$
\text{F1-score} = 2 \times \frac{\text{precision} \times \text{recall}}{\text{precision} + \text{recall}}.
\tag{5}
$$

Figure 6 is a calculation example of the IoU, precision, and recall. The presence of noise or other disturbances in the spectrum is not included. In the figure, serial numbers 1, 2, and 3 represent the three signals in the spectrum, and blue, orange, and green are used as marks respectively. The ground truth value is in the upper half, and the network prediction value is in the lower half. It can be seen that No. 1 and No. 2 are successfully predicted in the network prediction values, and their IoU values (0.772 and 1) both exceed the set threshold of 0.65. Thus, the $TP$ is 2. For signal No. 3, there is no eligible predicted value. In addition, there is one $FP$ between the No. 1 and No. 2 predictions, and the last prediction of the spectrum is also determined to be an $FP$ because $IoU_3$ is smaller than the threshold value. Thus, $FP$ is 2, and $FN$ is 1.

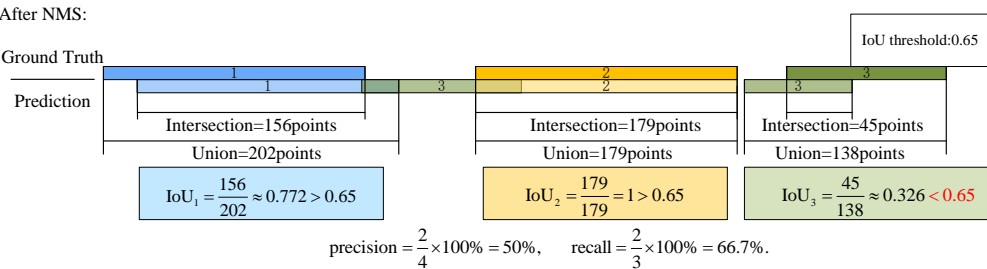

**Figure 6.** An example for IoU, precision, and recall.

As illustrated in the above process, the two indicators of precision and recall reflect the accuracy and completeness of the network, respectively. However, only considering the precision or recall is not a comprehensive consideration for evaluating model performance. Therefore, we also employ the F1-score, as it can also consider when it is difficult to fix the precision or recall.

The precision–recall curve (P-R curve) reflects the balance between precision and recall. Taking recall as the x-axis and accuracy as the y-axis, the curve changes as the confidence threshold fluctuates. The closer the P-R curve is to the upper right corner, the better the network performance.

### 3.4. Decision Fusion

Ensemble learning is a joint concept that improves the overall performance of model prediction by comprehensively considering multiple results after training multiple models. Many ensemble learning methods have been proposed. In this paper, we refer to the classical bagging method, which is representative of parallel ensemble learning. Bagging is based on the idea that a stable and reliable strong classifier can be obtained by combining several weak classifiers. The subsets mentioned in Algorithm 1 are for the bagging models. By transferring this idea to our scenario, each node will be a relatively weak sensing node, and a multi-receiver architecture will be a strong sensing system. The execution of multiple nodes is in parallel. An overview of the bagging weak models' training process is provided in Algorithm 2.

---

**Algorithm 2** Training the weak networks

---

**Input:** $N$ Multi-signal Training Samples $\{(x_i,\ y_i)\}_{i=1}^{N}$ in Dataset $D_2$;
**Output:** Weak Networks Group Parameters $\theta_M$
1:　TRAIN_MULTI($D_2$) :
2:　　　for $i \in$ Maxnodenumber $M$ **do**:
3:　　　　　$\{(x_i,\ y_i)\}_{i=1}^{n} \leftarrow$ $n$ random samples from $D_2$
4:　　　　　$\theta_{s_i} \leftarrow$ TRAIN_SN($\theta_{\text{FE}}$, $\{(x_i,\ y_i)\}_{i=1}^{n}$)
5:　　　**return** $\theta_M \leftarrow \{\theta_{S_1},\ \dots,\ \theta_{S_M}\}$

---

Weak models are voters, and this joint voting system is a strong model. Because of the task of this paper, we cannot use the maximum or average to calculate the joint predictions. Therefore, a threshold must be manually chosen to judge the validity of the prediction results. The judging procedure is shown in Algorithm 3.

As it is only effective to fuse multiple results by ensuring that each node is independent and diverse, we divide the original dataset into multiple subsets. Each subset is of the exact specification, and all are derived from the sampling of the original dataset. Subsets are allowed to overlap. To allow independence and diversity between single-node models, the nodes are trained on these subsets rather than the entire training set. In the final decision, a single node makes a prediction independently, which is sent to the fusion center. The best collective prediction is then made through an improved voting scheme, which is described as follows:

1.  Find all predictions of all single nodes and store them in the same section of spectrum.
2.  Divide the synthesized spectrum into 16 sections.
3.  Accumulate the number of center frequency points in each section. If the *C* category is higher than the certain voting threshold, it is counted as a joint decision result.
4.  Return to the original predictions, identify those that match the joint decision results, count their start and end positions in the spectrum data, and average them.

---

**Algorithm 3** Judging the joint decision results

---

**Input:** Prediction subsections, voting threshold $\phi$;
**Output:** Joint decision results;
1: **for** i $\in$ Subsections **do**:
2:     for $j \in$ Maxnodenumber $M$ **do**:
3:         $\text{class}_{ij}$, $\text{pos}_{ij} \leftarrow \text{Model}_j(\boldsymbol{\theta}_{s_i}, \text{Subsection}_i)$
4:     **if** $\sum_j \text{class}_{ij} \geq \phi$:
5:         class result$_i \leftarrow$ current class
6:         position result$_i \leftarrow$ average position
7: **return** class result, position result

---

It is essential to select an appropriate threshold. One that is too small will lead to an excessively aggressive prediction, while one that is too large will lead to results that are similar to a single node and cannot reflect the advantages of fusion. In this paper, we select and compare suitable voting threshold three after testing.

## 4. Experiments and Results

Variables, including the SNR, the quantity of signals, the modulation types of signals, and the number of nodes, are considered in this section. We employ the precision, recall, and F1-score to measure the models' performance, and precision–recall curves of the three models are drawn. Finally, we simulate the same multi-signal AMR by multiple nodes in different locations and compare it with the recognition performed by a single node. Unless otherwise specified, the three single-node recognition network models use the whole training set for 30 epochs, and the multi-node networks use 60% of the randomly selected training sets for 20 epochs.

### 4.1. Performance under Different SNRs

Figure 7 shows the results of three different single-node models under SNRs beyond the training set range. The spectrum data are mixed with different signal quantities, and noise with different energy is applied. Finally, the dataset is separated by the SNR. These single-node models are trained using the whole training set with 30 epochs. The performance of feature extraction using complex-conv is the best, and the overall performance of the other two networks shows little difference. In Table 4, the precision, recall, and F1-scores under some typical SNRs are shown in detail. It can be seen that the models are sensitive to noise. When the SNR drops to a certain low value, the performance results of the three networks have almost no difference. When SNR = 0 dB, the F1-scores of all networks are above 0.80. When SNR is less than $-14$ dB, the F1-scores are all below 0.20. Thus, all networks are hard to work with, or there are so many false alarms or misses that the prediction results are unusable.

### 4.2. Performance under Different Quantities of Signals

We consider different quantities of signals, where the test set is divided into four parts, and recall is the primary metric. A set of thresholds is selected, the precision is maintained above 90%, and the recall is tested. As illustrated in Figure 8, the quantities of signals influence the performance of the neural networks.

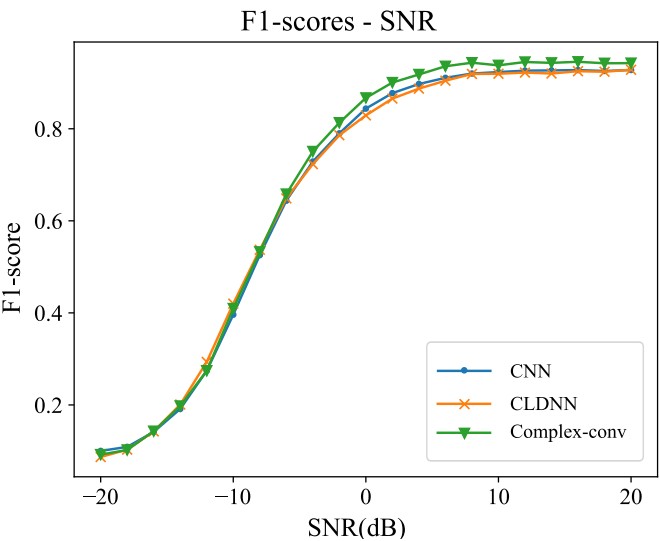

**Figure 7.** F1-scores for three models.

**Table 4.** Precision and recall (SNR = ±10 dB, ±6 dB, 0 dB, and −18 dB).

| | Precision | | | Recall | | | F1-Score | | |
|---|---|---|---|---|---|---|---|---|---|
| SNR(dB) | Complex-conv | CLDNN | CNN | Complex-conv | CLDNN | CNN | Complex-conv | CLDNN | CNN |
| 10 | 0.96205 | 0.93756 | 0.93991 | 0.91481 | 0.90221 | 0.90711 | 0.93784 | 0.91954 | 0.92322 |
| 6 | 0.95905 | 0.92890 | 0.93511 | 0.91091 | 0.88041 | 0.89481 | 0.93436 | 0.90400 | 0.91452 |
| 0 | 0.91343 | 0.89534 | 0.87564 | 0.81551 | 0.78221 | 0.79981 | 0.86170 | 0.82629 | 0.84488 |
| −6 | 0.74680 | 0.70079 | 0.72603 | 0.59871 | 0.58831 | 0.59371 | 0.66461 | 0.64996 | 0.64282 |
| −10 | 0.44593 | 0.45719 | 0.39159 | 0.37851 | 0.38861 | 0.40071 | 0.40946 | 0.42012 | 0.39610 |
| −18 | 0.09940 | 0.10380 | 0.08841 | 0.10501 | 0.10211 | 0.14161 | 0.10213 | 0.10295 | 0.10885 |

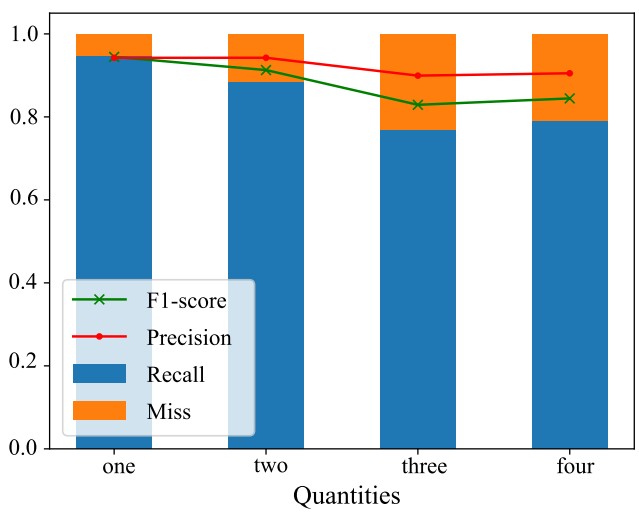

**Figure 8.** Recall and miss of the models when the precision remains above 90%.

In general, the fewer signals in the spectrum data, the better the models' performance. Figure 8 shows that the recall of the model is more sensitive to the number of signals in spectrum data, while the precision does not change as much. As the quantities of signals increase, there will be fewer free segments in the spectrum, and the network will miss signals with lower confidence when analyzing the crowded spectrum. The results in Figure 8 indicate that the difficulty of sensing different sources is varied. When choosing the network parameters, it is likely to conform to most types of signals but the confidence of the more difficult ones (such as SSB) is low.

### 4.3. Performance under Different Types of Signals

The performance of three networks for five types of signal types and mixtures is shown in Figure 9, where the final figure is the comparison values. An individual signal dataset is created, and there are $200 \times 21$ pieces of signal spectrum data of each type. The range of SNR of the test set exceeds training set one, ranging from $-20$ to 20 dB, with an interval of 2 dB. In most cases, the performance of the three models is not significantly different. Amongst these modulation types, the recognition performance of FSK signals demonstrates the highest consistency across different models. However, when the SSB signal is chosen as the communication modulation type, the performance of complex-conv as the feature extraction network is obviously better than other networks. This may be because the features of the SSB signal are significantly different in length from other signals, which means it is not well characterized. As a result, they have low recall, which affects the overall performance.

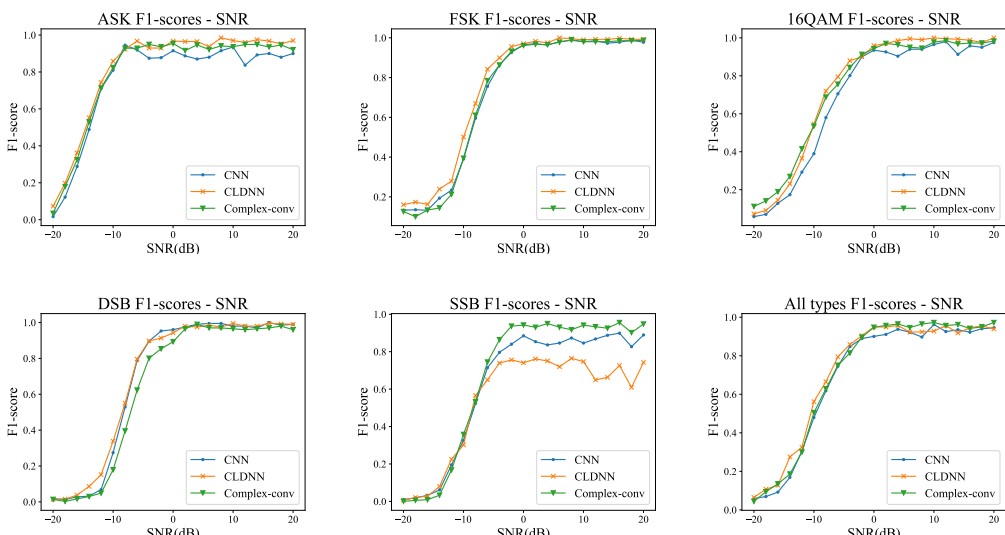

**Figure 9.** Performance under different types of signals.

### 4.4. Performance of the Fusion Method

The predictions from multiple nodes at different channels in the fusion center are fused by our method proposed in Section 3.4. Random noise with different energy is then applied to the spectrum dataset to simulate the different channels and positions for different nodes. The SNR of the spectrum dataset is controlled between $-8 \sim 6$ dB, and six different SNRs are randomly selected as the interference selections for the test data.

The average performance of the single node is the baseline for the fusion experiment. In addition, the experimental data after fusion are compared with the data when nodes = 6, according to the literature [36] in Table 5. The results of the fusion and baseline methods are shown in Table 6 and indicate that the fusion method performs better than single-node measurements. When using single-node measurements, it is clear that the training epochs and the size of the training dataset also significantly affect the model's performance. As shown in Table 5, when the number of nodes is six, the performance after fusion is slightly better than that of DAG-SVM.

### 4.5. Precision-Recall Curves

While a model cannot be fully measured by the precision or recall corresponding to a certain point, the overall performance of the P-R curve can provide a more comprehensive evaluation. The entire P-R curve is generated by changing the confidence threshold from high to low, where points close to the vertical axis have a larger confidence threshold and vice versa. P-R curves under different SNRs are shown in Figure 10, illustrating that the complex-conv model performs better in most situations.

**Table 5.** Fusion method performance under Complex-conv and DAG-SVM [36].

| Fusion and Conditions | Node = 1, Complex-conv with 20 Epochs, 60% Dataset | Node = 1, Complex-conv with 30 Epochs, 100% Dataset | Node = 6, Complex-conv with 20 Epochs, 60% Dataset | Node = 6, DAG-SVM [36] with Obtained Training Data Size |
| --- | --- | --- | --- | --- |
| F1-score (Sensing Accuracy) | 0.8405 | 0.8678 | **0.8933** | 0.8343 |

**Table 6.** Fusion method performance comparison with the single node.

| Feature Extraction Parts | Single | | | | | | Fusion | | |
| --- | --- | --- | --- | --- | --- | --- | --- | --- | --- |
| | 20 Epochs, 60% Dataset | | | 30 Epochs, 100% Dataset | | | 6 Nodes, 20 Epochs, 60% Dataset | | |
| | Precision | Recall | F1-Score | Precision | Recall | F1-Score | Precision | Recall | F1-Score |
| CNN | 0.79664 | 0.69881 | 0.74452 | 0.89661 | 0.80121 | 0.84623 | 0.91542 | 0.85281 | 0.88301 |
| CLDNN | 0.73912 | 0.61761 | 0.67292 | 0.88575 | 0.80001 | 0.84070 | 0.92616 | 0.86281 | 0.89336 |
| Complex-conv | 0.88570 | 0.79961 | 0.84046 | 0.91790 | 0.82281 | 0.86776 | 0.92503 | 0.86361 | 0.89327 |

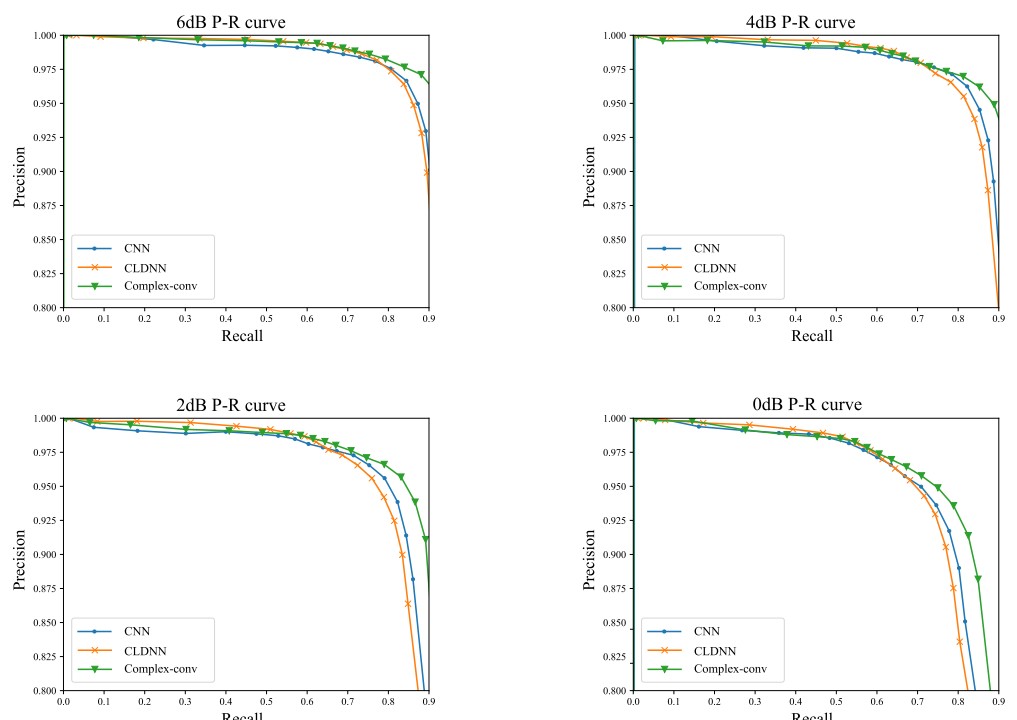

**Figure 10.** Performance under different types of signals.

*4.6. Model Runtime Duration*

The average inference time of the model is shown in Table 7. The duration counts the average duration from inputting a frequency domain sequence into the model until the prediction is completed during the test. CLDNN has a higher running time than the other two due to adding the LSTM layer.

**Table 7.** Average model runtime.

| Feature Extraction Parts | CNN | CLDNN | Complex-Conv |
| --- | --- | --- | --- |
| Runtime (ms) | 4.0298 | 21.0138 | 13.9534 |

## 5. Conclusions

This work proposed a multi-signal AMR method based on deep learning models and the fusion method to rationalize the monitoring and managing of electromagnetic spectrum resources. The radio monitoring method can monitor multiple signals on the spectrum simultaneously and return the offset frequency and occupied bandwidth of the signal relative to the scanning center frequency. Moreover, it can lay the foundation for subsequent judgments on whether it is non-cooperative interference and interference countermeasures, and contribute to the security of the communication environment. The model could work under multi-receiver scenarios with a good balance between complexity, speed, and energy consumption. End-to-end deep learning models can provide a more straightforward structure for radio monitoring. The architecture using the fusion method could also fuse more information. The decision-level fusion by the fusion center was enhanced by the diversity and independence of the slightly different models employed for prediction.

The proposed technique was tested by comparing the single-node and multi-node model. The single-node model achieved the highest F1-score under SNR = 10 dB, which was about 93.784%. When FSK was selected as the signal source, its recognition performance remained the most stable across different models. Performance results were higher when the fusion method employed multiple nodes rather than a single node. Using six models trained for 20 epochs and employing only 60% of the dataset for decision-level fusion could achieve better results than a single node trained for 30 epochs on the entire dataset. The results indicated that the proposed method is suitable for a distributed architecture. Using the method based on deep learning could improve the performance of radio monitoring, thus enhancing spectrum resource utilization efficiency.

The proposed method still has limitations. Feature extraction networks do not work effectively due to the similarity between modulation signals, and some specific sources, such as SSB, are not well sensing in the models. In addition, this method does not consider the security of the deep learning model itself in the non-cooperative adversarial communication scenarios [37], commonly observed in UAV communication scenarios. As the CNN model has been attacked by some methods in the recognition process [38], the security of the proposed method should also be strengthened. Due to the differences between the actual and simulated electromagnetic environments, future research directions may include testing with real-world signal datasets from UAV communication environments to supplement the analysis.

**Author Contributions:** Conceptualization, C.H.; Data curation, D.F.; Formal analysis, C.H. and D.F.; Funding acquisition, C.H.; Investigation, C.H. and Z.Z.; Methodology, C.H. and D.F.; Project administration, C.H.; Resources, C.H.; Software, D.F. and Z.Z.; Supervision, C.H.; Validation, C.H. and X.W.; Visualization, X.W.; Writing—original draft, D.F.; Writing—review and editing, C.H. and D.F. All authors have read and agreed to the published version of the manuscript.

**Funding:** This work is supported by the National Natural Science Foundation of China (62001137); the Fundamental Research Funds for the Central Universities (KY10800230034).

**Data Availability Statement:** The data are not publicly available due to the privacy of the research. However, the generation method and parameters of the dataset have been introduced in the manuscript, which can be reproduced using open-source tools.

**Conflicts of Interest:** The authors declare no conflict of interest.

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
