# Peer review of "A Deep Learning-Based Multi-Signal Radio Spectrum Monitoring Method for UAV Communication"

_drones, doi:10.3390/drones7080511_

Round 1

Reviewer 1 Report

The paper presents interesting ideas on Automatic Modulation Recognition system based na deep machine learning. The strongest aspect of the paper is the multisensor approach, where individual (sometimes weak) decisions are combined to form a more reliable decision. The paper is well structured and the ideas are clearly explained. The results are presented in a readable form and the corresponding discussion is satisfactory.

The quality of English langauge is generally fine, there are minor mistakes and errors which should be corrected.

Reviewer 2 Report

This paper proposed a deep Learning-based multi-signal radio spectrum monitoring method for UAV communications. Both the novelty and the presentation are fine. Some comments are given to improve this paper. (1) There are some format problems, such as  UAVs[1–4].->UAVs [1–4]. or fading effects[19,20].->or fading effects [19,20].UAVs first shown in the introduction part, it is better to give a full name. (2) Some related works are not discussed. The revised paper should be discussed in some related works of the field of communication and drones. (3) Some equation format should be revised as mathematic symbols, such as -14 dB,->$-14$ dB, 18dB,->18 dB,

The quality of this English writing is fine.

Reviewer 3 Report

The paper is interesting. Overall, it presents a good contribution supported by analyses and figures. However, there are some issues to be addressed.

1- Justifying the novelty of the proposed work could be better supported by presenting a more detailed comparison with the related works (e.g. a table could be helpful considering several factors such as the artificial intelligence technique used, deep learning or swarm intelligence, or even hybridized methods, and other comparison factors).

2- Although the efficiency of the method has been claimed, it seems the evaluation metrics employed are for the effectiveness of the method. I did not see a discussion of the method's efficiency in terms of its complexity or the time needed for signal identification. There should be more analyses and discussions on these aspects.

3- In the conclusion, it is mentioned: "This radio monitoring method can contribute to improving the quality and security of UAV communication". I do not see any discussion or analysis on how the method can improve UAV communication security. 

The English language used in the paper has minor issues such as: "This paper proposed a method that utilizes..." should be "This paper proposes..."

Reviewer 4 Report

This paper describes an automatic modulation recognition technique using deep learning technique to identify multiple signals for MIMO receivers. At first, a spectrum dataset is obtained and a label is developed. Then, feature extraction (CNN, CLDNN, Complex-Conv) and detection networks are deployed to extract the presence and location information of signals in the spectrum. Finally, decision-level fusion is used to combine the output results of multiple nodes. Five modulation schemes namely ASK, FSK, 16QAM, DSB-SC and SSB is used in the experiment with different SNRs. FSK performed the best as per the results obtained. It is expected that the proposed scheme can be applied for wireless radio spectrum monitoring in complex environments.

It is not a correct claim that previous works in AMD never discussed multiple modulations. You can find such work in literature such as:

Wang, Y., Wang, J., Zhang, W., Yang, J., & Gui, G. (2020). Deep learning-based cooperative automatic modulation classification method for MIMO systems. Ieee transactions on vehicular technology, 69(4), 4575-4579.

Wang, Y., Gui, J., Yin, Y., Wang, J., Sun, J., Gui, G., ... & Adachi, F. (2020). Automatic modulation classification for MIMO systems via deep learning and zero-forcing equalization. IEEE transactions on vehicular technology, 69(5), 5688-5692.

Bouchenak, S., Merzougui, R., Harrou, F., Dairi, A., & Sun, Y. (2022). A semi-supervised modulation identification in MIMO systems: A deep learning strategy. IEEE Access, 10, 76622-76635.

Shi, J., Hong, S., Cai, C., Wang, Y., Huang, H., & Gui, G. (2020). Deep learning-based automatic modulation recognition method in the presence of phase offset. IEEE Access, 8, 42841-42847.

* Abstract needs revison for coherence with the paper objectives.

* Several typos in the text that need correction. For example:

Table 5 caption: 'Comparation' should be 'Comparison'.

Line 436: This sentence needs revision 'An end-to-end deep learning models were employed in nodes, which ensured the radio monitoring had low complexity and fast response speeds.'

The paper must be revised by a technical English expert for rephrasing major parts of the paper including abstract and conclusion.

Why FSK performance is highest? Please explain.

Why SSB performance is lowest in Fig. 9? Please explain.

Orthogonal time-frequency space (OTFS) modulation for UAV communication is popular these days due to the noise suppression properties. Is it possible to include it in this research?

For the UAV communication, the topology changes frequently, hence link breakages affects the communication. Moreover, there are other issues as well, such as routing issues, mobility issues, scalability issues, reliability issue, power issues due to battery limitations etc. These are not highlighted in the text.

It is important to test the proposed technique in HIL on a real hardware e.g. using Software defined radio (SDR) to assess the quality of classification in real time? This will verify the simulated results.

English language revision is required.

Author Response

Please see the attachment. A more specific language revision will be shown in the latest manuscript. Thanks again for your valuable comments. 

Round 2

Reviewer 4 Report

The paper has been revised to incorporate necessary changes. Therefore, i have no further suggestions.